# Dietary patterns; serum concentrations of selenium, copper, and zinc; copper/zinc ratio; and total antioxidant status in patients with glaucoma

Izabela Zawadzka[1], Maryla Młynarczyk[1], Martyna Falkowska[2], Katarzyna Socha[2], Joanna Konopińska[1]*

1 Department of Ophthalmology, Medical University of Białystok, Białystok, Poland, 2 Faculty of Pharmacy with the Division of Laboratory Medicine, Department of Bromatology, Medical University of Białystok, Białystok, Poland

* joannakonopinska@o2.pl

## Abstract

This study aimed to identify the biochemical parameters that determine the occurrence of glaucoma and assess the correlation between oxidative stress and clinical data in patients with glaucoma and healthy controls. We enrolled 169 participants; the glaucoma group comprised 104 patients with primary open-angle, pseudoexfoliation, or angle-closure glaucoma, and the control group comprised 65 healthy individuals. Serum concentrations of selenium (Se), copper (Cu), and zinc (Zn); Cu/Zn ratio; and total antioxidant status were measured in both groups. Significantly lower Se and Zn serum levels were observed in men (67.7 ± 17.14 g/L and 0.76 ± 0.11 mg/L, respectively) and women (68.73 ± 16.21 g/L and 0.76 ± 0.13, respectively) with glaucoma. Moreover, significant correlations were identified between serum Se concentration and corrected distance visual acuity (CDVA) and between serum Cu concentration and CDVA ($p < 0.005$ and $p < 0.05$, respectively). We also observed a significant positive correlation ($r = 0.244$, $p < 0.05$) between pRNFL thickness and BMI and a negative correlation ($r = -0.289$, $p < 0.05$) between serum Se concentration and the age of male patients with glaucoma. Additionally, the percentages of participants with below-normal, normal, and above-normal Se, Zn, and total antioxidant capacity serum levels were compared between both groups. Compared with healthy controls, a significantly higher percentage of patients with glaucoma had a below-normal Se serum concentration. A notable negative correlation was observed between Zn and copper serum levels of patients with glaucoma in both sexes. We believe that this study serves as a basis for considering personalized nutritional therapy for the prevention and supportive treatment of patients with glaucoma.

**Data Availability Statement:** All relevant data are within the manuscript and its Supporting information files.

## Introduction

Glaucoma is characterized by progressive optic nerve degeneration which has multiple contributing factors. This pathology results in a gradual loss of retinal ganglion cells (RGCs) and

**Funding:** The author(s) received no specific funding for this work.

**Competing interests:** No authors have competing interest.

associated axons, leading to distinct changes in the optic disc and the retinal nerve fiber layer (RNFL), ultimately causing vision loss. Glaucoma affects approximately 3.54% (2.09–5.82%) of people aged 40–80 years [1–3], with projections of 111.8 million cases by 2040. Despite advanced therapeutic methods, such as pharmacology, lasers, and surgery, glaucoma remains globally the first leading cause of irreversible blindness. The only modifiable risk factor for glaucoma is elevated intraocular pressure (IOP); nonetheless, glaucoma can develop in individuals with normal IOP. Moreover, glaucomatous optic neuropathy in some patients progresses despite administering IOP-lowering drugs or surgery. This might indicate the presence of undefined molecular mechanisms underlying glaucoma development. The pathogenesis of glaucoma is associated with oxidative stress (OS), resulting from an imbalance between the synthesis of reactive oxygen species (ROS) and the capacity of antioxidant and antioxidant-repairing systems. Deficiency or over supplementation of some macro- and microelements also adversely affects the course of glaucomatous optic neuropathy [3]. Given the multifactorial etiology of glaucoma and its chronic nature, an appropriate diet and adjustment of the dietary intake of selected nutrients should be considered as adjuncts to treatment that may inhibit the disease progression. Antioxidants, including enzymatic cofactors, such as Se, Cu, and Zn, serve as antioxidant defense markers. Moreover, metal ions, such as Fe and Zn, are vital for normal cellular function, particularly in nervous system synapses. Fe and Zn are the key cofactors in neurotransmitter synthesis [4, 5]. Zn modulates synaptic activity and serves as an intracellular transmitter. Additionally, free forms of metals and metal ions are transduced by metalloproteins that influence specific intracellular functions. The involvement of these ions in the pathogenesis of neurodegenerative diseases, such as Parkinson's disease and Alzheimer's disease, is of growing interest as a potential target for novel therapies [6]. The concentrations of metal ions in the retina, aqueous humor, and blood serum have been analyzed in glaucoma and other neurodegenerative diseases in laboratory animals and humans. In the retinas of mice with glaucoma, the $Fe^+$ concentration is decreased, whereas that of Zn is twice as high as that in healthy animals. An elevated blood concentration of Fe in humans is associated with an increased glaucoma risk [7, 8]. Another study assessed Se, Cu, and Zn levels, as well as Cu/Zn and Cu/Se ratios, in patients with acute ischemic stroke and healthy controls [9] and found that patients with acute ischemic stroke presented with markedly lowered Se and Zn levels and elevated Cu/Zn and Cu/Se ratios. Such alterations in trace element content may cause inflammation and OS, eventually leading to stroke [9]. In a different study [5], lower serum concentrations of Zn and higher Cu/Zn ratios in patients with multiple sclerosis implied an association with OS. Hence, dietary supplementation with Zn may benefit patients with multiple sclerosis [10]. We speculate that a similar mechanism may be involved in the pathogenesis of glaucoma, another neurodegenerative disease. Evidence suggests that OS is crucial in the pathogenesis of glaucoma [5]. Therefore, identifying the disruptions of pro-oxidants in the antioxidant balance specific to the various types of glaucoma is crucial. This study aimed to identify biochemical parameters that determine an increased risk of glaucoma occurrence. Moreover, the correlations between OS and the clinical data of patients with glaucoma and those of healthy controls were analyzed. To the best of our knowledge, this is the first study to investigate trace mineral status in patients with glaucoma in a Polish population.

## Materials and methods

### Inclusion and exclusion criteria

The study data were gathered following approval from the Bioethics Committee of the Medical University of Bialystok (approval number: APK.002.174.2022), per the principles outlined in the Declaration of Helsinki, and in compliance with good medical practice. All participants

provided written informed consent for the use of their blood test results, nutritional survey data, and examination findings in scientific research publications. The recruitment period began on February 24, 2022, and ended on December 12, 2022.

We enrolled 169 participants; the glaucoma group consisted of 104 patients with glaucoma, and the healthy control group included 65 individuals. Patients with glaucoma were recruited from the Ophthalmology Clinic of the Medical University of Bialystok, a tertiary center for patients with glaucoma from the Podlaskie Voivodeship. Control group was recruited from patient's family members attending them for the control visit as well as clinical personnel. All individuals underwent ophthalmological examinations, which included a slit-lamp examination, corrected distance visual acuity (CDVA) assessment, IOP measurement using a Goldman applanation tonometer or a portable Tonopen device (in the case of patients with disabilities), and fundoscopy. The anterior eye segment and iridocorneal angle were examined in detail to accurately identify the glaucoma type (primary open-angle glaucoma [POAG], pseudoexfoliation glaucoma [PEXG], or primary angle-closure glaucoma [PACG]) in each patient. The severity of glaucoma was determined by optic nerve disc evaluation, the presence of optic disc cupping, focal damage, and the 30–2˚ Humphrey VF test (Carl Zeiss Meditec Dublin, CA), according to Hodapp's classification. Optical coherence tomography (Heidelberg Engineering) was performed to measure the peripapillary RNFL (pRNFL).

The control group consisted of 65 individuals without any ophthalmic disease, comprising cataract. The exclusion criteria for the study included a history of severe systemic illnesses, such as liver and intestinal diseases resulting in malabsorption syndrome, hypothyroidism, hypovitaminosis, psychiatric disorders, autoimmune diseases, or neoplasms. Individuals with a history of alcohol abuse syndrome, prior ophthalmological surgery, ocular inflammation, retinal vascular diseases, or diabetic retinopathy were excluded from the study. Additionally, individuals taking dietary vitamin supplements, especially those containing Se, Zn, or Cu, or medications that could potentially affect the trace element levels were excluded from the study.

## Data collection, sample preparation, and biochemical analysis

Dietary data were collected using a food frequency questionnaire developed by the Committee of Human Nutrition Science of the Polish Academy of Sciences. Patients diagnosed with glaucoma were asked to complete a questionnaire assessing the frequency of their consumption of various food products. Completion of these questionnaires was overseen by a currently attending primary physician in the ophthalmology department. The questionnaire included 34 distinct groups of food items: white bread, wholegrain bread, sweet bread, rice, oatmeal, groats, milk, white cheese, poultry, offal, sausages, ham, bacon, lard, meat, tinned meat, fish, eggs, butter, margarine, oils, potatoes, raw vegetables, fruit, sugar, honey, marmalade, sweet drinks, legumes, beer, wine, vodka, coffee, and tea. For this study, consumption was frequent if specific food products were consumed twice to thrice weekly or more, except for fish and fish products, for which consumption once or more weekly was considered frequent. Consumption was considered seldom if food items were consumed not more than once weekly. This categorization aligns with the recommendations of the Committee on Human Nutrition Science of the Polish Academy of Sciences for assessing the frequency of consuming specific food products.

Blood samples (approximately 6 mL) were collected from the study participants using vacutainer test tubes containing a clot activator recommended for elemental analysis (Becton Dickinson, France). These collected samples were centrifuged for 10 min at approximately 1000 × *g*. Subsequently, sera were obtained and preserved by freezing at −20˚C. Deproteination was performed using 1 mol/L spectral-grade nitric acid (Merck, Germany) to prepare the serum

samples for analysis. Additionally, 1% Triton X-100 was introduced into the samples and vortexed, and another round of centrifugation was performed, lasting 10 min. The Zn concentration was quantified in the resulting supernatant, whereas Cu content was determined after suitable dilution in 0.1 mol/L nitric acid. Se concentrations were determined directly after a 1:1 dilution with 0.2% Triton X-100.

To analyze the influence of the frequency of consuming different food groups on the concentration of a selected element, we used the stepwise forward multiple regression method.

## Measurement of mineral components and total antioxidant status

The serum concentrations of Se, Cu, and Zn were assessed using atomic absorption spectrometry techniques, specifically electrothermal for Se and Cu and flame for Zn, with Zeeman background correction. These measurements were carried out at wavelengths of 196 nm for Se, 324.8 nm for Cu, and 213.9 nm for Zn, utilizing the Z-2000 instrument (Hitachi, Tokyo, Japan). Calibration curves were constructed using working solutions derived from 1 g/L standard solutions obtained from Merck (Darmstadt, Germany). The limits of detection for Se, Cu, and Zn were 1.84 μg/L, 0.51 μg/L, and 0.011 mg/L, respectively. A certified matrix reference material in the form of human serum (Seronorm Trace Elements, Serum Level 1, 0903106; Sero AS, Billingstad, Norway) was employed to ensure the accuracy and precision of the analytical methods. The results obtained for the control samples were aligned with the established reference values. The analytical methods for Se, Cu, and Zn had precision values of 3.3%, 2.3%, and 1.8%, respectively.

The Department of Bromatology at the Medical University of Bialystok actively participated in a quality control program for trace element analysis under the supervision of the National Institute of Public Health, the National Institute of Hygiene, and the Institute of Nuclear Chemistry and Physics, all in Warsaw, Poland. The serum concentrations of Se, Cu, and Zn in the glaucoma group were subsequently evaluated in comparison to the reference values of healthy controls (66–104 μg/L, 0.7–1.6 mg/L, and 0.7–1.3 mg/L, respectively) [11]. The Cu/Zn molar ratio was calculated using Microsoft Excel (Microsoft Corporation, Redmond, Washington, USA).

The total antioxidant status (TAS) of the serum was determined spectrophotometrically at 600 nm. The analysis was conducted using commercially available test kits provided by Randox Laboratories Ltd. (Crumlin, UK) and a UV–Vis spectrophotometer (Cintra 3030; GBC, Melbourne, Australia). The assay involved incubating ABTS® (2,20-Azino-di-[3-ethyl-benzthiazoline sulphonate]) with peroxidase (metmyoglobin) and $H_2O_2$ to generate the radical cation ABTS®*+. This radical exhibited a relatively stable blue-green color, which was measured at 600 nm. The antioxidants in the tested samples caused a reduction in color production—proportional to their concentration. The accuracy of this method was validated using a TAS Control kit from Randox Laboratories, Ltd. The reference range for TAS in the serum was 1.3–1.77 mmol/L [12].

## Statistical analysis

Statistical analyses were conducted using the Statistica software (version 13.0; TIBCO Software Inc., Palo Alto, CA, USA). Data normality was assessed using the Kolmogorov–Smirnov and Shapiro–Wilk tests. Variations between independent groups were evaluated using the Mann–Whitney U or ANOVA Kruskal–Wallis test, depending on the number of groups involved. Group comparisons were performed using Pearson's chi-square test and Fisher's exact test. Correlations were determined using Spearman's rank test. Multiple stepwise regression

analysis was employed to assess the impact of dietary habits on Se, Cu, and Zn levels and TAS in the study participants. Statistical significance was set at $p < 0.05$ (S2 File).

## Results

### Demographic characteristics of the study population

The analysis was based on patients with glaucoma (n = 104) and control participants (n = 65). The percentages of men were 32.7% and 23.1%, whereas those of women were 67.3% and 76.9%, respectively, with no significant differences between the study groups. The mean age in the glaucoma group was 70.4 ± 10.6 years, and that in the control group was 62.9 ± 11.8 years. No significant difference in age existed between the groups. Body mass index (BMI) was available only for patients with glaucoma, which was 28.46 ± 4.42 kg/m² (Table 1).

### Se, Cu, Zn, and TAS levels and Cu/Zn ratio

In our study, significant differences were observed in the serum levels of Se, Zn, and the Cu/Zn ratio between the glaucoma and control groups ($p < 0.001$, $p < 0.001$, and $p = 0.008$, respectively). The serum Se concentration was significantly lower in patients with glaucoma (68.40 ± 16.44 µg/L) than in healthy individuals (86.02 ± 50.04 µg/L). In addition, Zn levels were significantly lower in the glaucoma group (0.76 ± 0.12 mg/L) than in the control group (0.85 ± 0.16 mg/L). Another significantly different parameter was the Cu/Zn ratio, which differed between the glaucoma and control groups (1.51 ± 0.52 and 1.35 ± 0.57, respectively). The Cu and TAS levels did not significantly differ between groups (Table 2); however, we observed a negative correlation (r = -0.396, $p < 0.05$) between these parameters.

**Table 1. Demographic characteristics of study groups.**

| Variable | Glaucoma group | | Control group | |
|---|---|---|---|---|
| | n (%) or mean ± SD | Median (Q1; Q3) | n (%) or mean ± SD | Median (Q1; Q3) |
| N | 104 (100.0) | - | 65 (100.0) | - |
| Sex, male | 34 (32.4) | - | 15 (23.1) | - |
| Sex, female | 70 (67.3) | - | 50 (76.9) | - |
| Age, years | 70.4 ± 10.6 | 73 (66; 77) | 62.9 ± 11.8 | 64 (55; 70) |
| BMI, kg/m² | 28.46 ± 4.42 | 27.73 (25.39; 31.04) | - | - |

Notes: BMI, body mass index; Q1, first quartile; Q3, third quartile; SD, standard deviation

**Table 2. Comparison of Se, Zn, Cu, and TAS concentrations and Cu/Zn ratio between the study groups.**

| Variable | Glaucoma group | | Control group | | p-value |
|---|---|---|---|---|---|
| | Mean ± SD | Median (Q1; Q3) | Mean ± SD | Median (Q1; Q3) | |
| Se, µg/L | 68.40 ± 16.44 | 66.57 (57.90; 77.23) | 86.02 ± 50.04 | 75.48 (69.07; 86.87) | **< 0.001*** |
| Zn, mg/L | 0.76 ± 0.12 | 0.76 (0.68; 0.82) | 0.85 ± 0.16 | 0.82 (0.75; 0.94) | **< 0.001*** |
| Cu, mg/L | 1.10 ± 0.34 | 1.09 (0.89; 1.29) | 1.08 ± 0.37 | 0.99 (0.85; 1.24) | 0.261 |
| Cu/Zn, molar ratio | 1.51 ± 0.52 | 1.47 (1.14; 1.81) | 1.35 ± 0.57 | 1.29 (0.98; 1.45) | **0.008*** |
| TAS, mmol/L | 1.51 ± 0.28 | 1.55 (1.41; 1.65) | 1.50 ± 0.58 | 1.50 (1.07; 1.79) | 0.408 |

Notes: Q1, first quartile; Q3, third quartile; SD, standard deviation; TAS, total antioxidant status

## Correlations of micronutrients with pRNFL thickness, CDVA, IOP, and BMI

Next, we investigated the correlations between micronutrients and pRNFL thickness, CDVA, IOP, BMI, glaucoma type, and sex within the study population Significant correlations were identified between serum Se concentration and CDVA and between serum Cu concentration and CDVA ($p < 0.005$ and $p < 0.05$, respectively). We also observed a significant positive correlation ($r = 0.244$, $p < 0.05$) between pRNFL thickness and BMI and a negative correlation ($r = -0.289$, $p < 0.05$) between serum Se concentration and the age of male patients with glaucoma.

Additionally, we assessed the levels of serum trace elements among individuals diagnosed with glaucoma, distinguishing between smokers and non-smokers. We identified a significant reduction in serum Se concentrations among smoking patients with glaucoma (62.48 ± 18.14 µg/L) compared to non-smoking patients with glaucoma (71.93 ± 16.07 µg/L). However, we did not detect differences in the serum concentrations of other microelements between these two groups (Table 3).

In the glaucoma group, sex-dependent variability in the serum concentrations of Se, Zn, and Cu and Cu/Zn ratios was also observed. Significantly lower serum levels of Se and Zn were observed in men (67.7 ± 17.14 µg/L and 0.76 ± 0.11 mg/L, respectively) and women (68.73 ± 16.21 µg/L and 0.76 ± 0.13 mg/L, respectively) diagnosed with glaucoma than in healthy men (78.64 ± 15.13 µg/L and 0.82 ± 0.21 mg/L, respectively) and women (88.23 ± 56.43 µg/L and 0.71 ± 0.14 mg/L, respectively). Moreover, decreased serum Cu concentrations were observed in men (1.07 ± 0.31 mg/L) and women (1.10 ± 0.35 mg/L) diagnosed with glaucoma compared to their counterparts in the control group (men: 1.14 ± 0.42 mg/L, women: 1.06 ± 0.36 mg/L; Table 4).

**Table 3. Serum Se, Zn, Cu, and TAS concentrations and Cu/Zn molar ratio in patients with glaucoma according to smoking status.**

| Variable | | Smokers (n = 25) | Non-smokers (n = 75) |
|---|---|---|---|
| Se, µg/L | Mean ± SD | 62.48 ± 18.14 | 71.93 ± 16.07 |
| | Min–max | 30.39–106.90 | 43.74–113.26 |
| | Median (Q1; Q3) | 61.87 (57.27; 71.39) | 70.25 (59.33; 83.11) |
| p-value | | < 0.001* | |
| Zn, mg/L | Mean ± SD | 0.793 ± 0.113 | 0.761 ± 0.117 |
| | Min–max | 0.541–1.024 | 0.476–1.061 |
| | Median (Q1; Q3) | 0.739 (0.723; 0.851) | 0.761 (0.679; 0.816) |
| p-value | | 0.151 | |
| Cu, mg/L | Mean ± SD | 1.096 ± 0.364 | 1.079 ± 0.346 |
| | Min–max | 0.407–1.711 | 0.511–2.041 |
| | Median (Q1; Q3) | 1.057 (0.947; 1.312) | 1.073 (0.857; 1.306) |
| p-value | | 0.787 | |
| Cu/Zn, molar ratio | Mean ± SD | 1.42 ± 0.44 | 1.49 ± 0.52 |
| | Min–max | 0.51–2.36 | 0.60–3.043 |
| | Median (Q1; Q3) | 1.38 (1.11; 1.71) | 1.46 (1.13; 1.78) |
| p-value | | 0.593 | |
| TAS, mmol/L | Mean ± SD | 1.52 ± 0.20 | 1.50 ± 0.26 |
| | Min–max | 1.27–1.94 | 0.38–2.11 |
| | Median (Q1; Q3) | 1.55 (1.41; 1.65) | 1.53 (1.41; 1.64) |
| p-value | | 0.859 | |

Notes: Q1, first quartile; Q3, third quartile; SD, standard deviation; TAS, total antioxidant status

**Table 4. Sex-dependent distribution of microelement concentrations and total antioxidant status.**

| Variable | | Patients with glaucoma | | Control group | |
|---|---|---|---|---|---|
| | | Men (A) | Women (B) | Men (C) | Women (D) |
| | | (n = 34) | (n = 70) | (n = 15) | (n = 50) |
| Se, µg/L | Mean ± SD | 67.7 ± 17.14 | 68.73 ± 16.21 | 78.64 ± 15.13 | 88.23 ± 56.43 |
| | Min–max | 35.05–104.92 | 30.39–113.26 | 50.88–106.8 | 48.63–388.67 |
| | Median (Q1; Q3) | 63.80 (56.65; 75.39) | 66.79 (58.98; 79.88) | 77.89 (75.54; 89.13) | 74.52 (68.24; 86.1) |
| **p-value** | | **A vs. C: 0.02\*, B vs. D: 0.004\*** | | | |
| Zn, mg/L | Mean ± SD | 0.76 ± 0.11 | 0.76 ± 0.13 | 0.82 ± 0.21 | 0.71 ± 0.14 |
| | Min–max | 0.56–1.10 | 0.48–1.06 | 0.68–1.50 | 0.55–1.12 |
| | Median (Q1; Q3) | 0.76 (0.69; 0.81) | 0.75 (0.67; 0.83) | 0.90 (0.76; 1.04) | 0.82 (0.74; 0.91) |
| **p-value** | | **A vs. C: 0.006\*, B vs. D: 0.003\*** | | | |
| Cu, mg/L | Mean ± SD | 1.07 ± 0.31 | 1.10 ± 0.35 | 1.14 ± 0.42 | 1.06 ± 0.36 |
| | Min–max | 0.52–1.75 | 0.41–2.04 | 0.70–2.36 | 0.53–2.31 |
| | Median (Q1; Q3) | 1.09 (0.88; 1.26) | 1.1 (0.90; 1.31) | 0.99 (0.86; 1.37) | 0.99 (0.81; 1.24) |
| **p-value** | | **A vs. C: 0.005\*, B vs. D: 0.004\*** | | | |
| Cu/Zn, molar ratio | Mean ± SD | 1.46 ± 0.45 | 1.54 ± 0.55 | 1.35 ± 0.68 | 1.35 ± 0.54 |
| | Min–max | 0.63–2.53 | 0.51–3.08 | 0.70–3.44 | 0.59–3.32 |
| | Median (Q1; Q3) | 1.43 (1.01; 1.76) | 1.47 (1.15; 1.83) | 1.25 (0.96; 1.44) | 1.29 (0.99; 1.47) |
| **p-value** | | A vs. C: ns, **B vs. D: 0.024\*** | | | |
| TAS, mmol/L | Mean ± SD | 1.57 ± 0.27 | 1.49 ± 0.28 | 1.50 ± 0.71 | 1.51 ± 0.55 |
| | Min–max | 1.01–2.11 | 0.38–2.01 | 0.87–3.47 | 0.63–3.50 |
| | Median (Q1; Q3) | 1.56 (1.43; 1.70) | 1.52 (1.35; 1.65) | 1.37 (1.01; 1.65) | 1.53 (1.09; 1.79) |
| **p-value** | | A vs. C: ns, B vs. D: = ns | | | |

Notes: ns, not significant; Q1, first quartile; Q3, third quartile; SD, standard deviation; TAS, total antioxidant status

Significant differences in the proportions of participants with Se, Zn, and TAS levels in relation to the standard values were observed between the two study groups ($p < 0.001$, $p < 0.02$, and $p < 0.001$, respectively). The proportions of participants with standard Se levels were 46.2% and 75.4% in the glaucoma and control groups, respectively. A higher proportion of participants had Se levels below the standard in the glaucoma group than in the control group (50.0% vs. 15.4%). The number of participants with Se levels above the standard was lower in the glaucoma group (3.8%, n = 4 vs. 9.2%, n = 6). The proportion of participants with standard Zn levels was lower among patients with glaucoma (64.4%) than among control participants (80.0%). A higher proportion of participants had Zn levels below the standard in the glaucoma group than in the control group (35.6% vs. 18.5%). No patient had Zn levels above the standard level in the glaucoma group. However, the Zn level exceeded the standard in one participant of the control group. The proportion of participants with standard TAS levels was over two-fold higher in the glaucoma group than in the control group (69.2% vs. 30.4%). A lower proportion of participants had TAS levels below the standard in the glaucoma group than in the control group (18.3% vs. 48.2%). The number of participants with TAS levels above the standard was also lower in the glaucoma group (12.5%) than in the control group (21.4%). No significant difference was confirmed in participants with below-the-standard/standard/above-the-standard Cu levels (Table 5).

Multiple stepwise regression analysis revealed that 15 dietary habits determined 50% of the serum Se concentration. An increase in Se concentration was significantly associated with the frequent consumption of fish, meat, coffee, milk, and eggs. In contrast, the frequent

**Table 5. Comparison of Se, Zn, Cu, and TAS levels between the study groups in relation to reference values.**

| Variable | Glaucoma group | Control group | p-value |
|---|---|---|---|
| **Se** | | | |
| Low | 52 (50.0) | 10 (15.4) | **< 0.001** |
| Normal | 48 (46.2) | 49 (75.4) | |
| High | 4 (3.8) | 6 (9.2) | |
| **Zn** | | | |
| Low | 37 (35.6) | 12 (18.5) | **0.014**[1] |
| Normal | 67 (64.4) | 52 (80.0) | |
| High | 0 (0.0) | 1 (1.5) | |
| **Cu** | | | |
| Low | 14 (13.4) | 4 (6.2) | 0.312 |
| Normal | 84 (80.8) | 56 (86.2) | |
| High | 6 (5.8) | 5 (7.7) | |
| **TAS** | | | |
| Low | 19 (18.3) | 27 (48.2) | **< 0.001** |
| Normal | 72 (69.2) | 17 (30.4) | |
| High | 13 (12.5) | 12 (21.4) | |

Notes: Data were compared using Pearson's chi-square test or Fisher's exact test[1]. TAS, total antioxidant status

consumption of beer, bacon, lard, white bread, and potatoes was associated with a decrease in Se levels. Serum Zn concentration was influenced by 15 dietary habits up to 42%, with positive effects associated with frequent consumption of ham, offal, potatoes, and milk, whereas vegetable oil and raw vegetables had a negative effect. According to the analysis, the consumption frequency of five selected food products explain 38% of the variation in Cu concentration in your study population. Bacon, lard, potatoes, grits, and rice had a negative effect; however, frequent consumption of fruits was associated with positive effects on serum Cu concentration. Moreover, serum TAS concentration was influenced by 12 dietary habits up to 13%, and frequent consumption of margarine significantly decreased serum TAS levels (S1 Table). We did not observe notable variations in the serum concentrations of these elements in patients with different types of glaucoma. The determined serum levels were not significantly different among patients with different types of glaucoma (Table 6).

**Table 6. Comparison of Se, Zn, Cu, and TAS levels and Cu/Zn ratio between glaucoma types.**

| Variable | POAG (n = 58) | | PEXG (n = 16) | | PACG (n = 17) | | Other types (n = 7) | | p |
|---|---|---|---|---|---|---|---|---|---|
| | Mean ± SD | Median (Q1; Q3) | Mean ± SD | Median (Q1; Q3) | Mean ± SD | Median (Q1; Q3) | Mean ± SD | Median (Q1; Q3) | |
| **Se, mg/L** | 66.44 ± 17.37 | 63.74 (55.42; 75.97) | 76.40 ± 18.53 | 71.05 (63.23; 81.46) | 72.73 ± 17.14 | 72.81 (59.88; 83.02) | 67.63 ± 11.38 | 66.91 (59.78; 76.19) | 0.175 |
| **Zn, mg/L** | 0.77 ± 0.14 | 0.76 (0.68; 0.85) | 0.74 ± 0.09 | 0.77 (0.70; 0.81) | 0.76 ± 0.09 | 0.76 (0.71; 0.82) | 0.77 ± 0.11 | 0.75 (0.71; 0.82) | 0.912 |
| **Cu, mg/L** | 1.05 ± 0.36 | 1.02 (0.83; 1.28) | 1.22 ± 0.32 | 1.17 (1.05; 1.45) | 1.03 ± 0.28 | 1.07 (0.92; 1.17) | 1.17 ± 0.30 | 1.16 (0.99; 1.26) | 0.281 |
| **Cu/Zn, molar ratio** | 1.45 ± 0.55 | 1.38 (1.02; 1.79) | 1.72 ± 0.56 | 1.61 (1.38; 1.94) | 1.42 ± 0.41 | 1.47 (1.20; 1.71) | 1.56 ± 0.35 | 1.67 (1.28; 1.79) | 0.263 |
| **TAS, mmol/L** | 1.53 ± 0.25 | 1.52 (1.40; 1.65) | 1.53 ± 0.28 | 1.55 (1.42; 1.64) | 1.49 ± 0.35 | 1.58 (1.43; 1.64) | 1.51 ± 0.17 | 1.50 (1.45; 1.65) | 0.977[1] |

Notes: POAG, primary open angle glaucoma; PEXG, pseudoexfoliation glaucoma; PACG, primary angle-closure glaucoma; Q1, first quartile; Q3, third quartile; SD, standard deviation. Data were compared with ANOVA or Kruskal–Wallis test[1].

**Table 7. Relationship between the frequency of fish consumption and the serum concentration of selenium in patients with glaucoma.**

| Fish consumption frequency | Number of patients with glaucoma | Concentration of selenium in serum (µg/L) | p |
|---|---|---|---|
| Frequent (more than once a week) | 51 | 74.92 ± 17.49 | < 0.005* |
| Seldom (less or equal to once a week) | 53 | 64.87 ± 13.40 | |

In the glaucoma group, the effect of fish consumption as a source of selenium was statistically significant; patients who consumed fish more frequently had higher selenium levels compared to those who consumed fish less frequently (without considering other dietary factors), Table 7.

## Discussion

In our research, study participants with glaucoma had considerably lower Se and Zn serum levels along with a noticeably higher Cu/Zn ratio than healthy study participants. Additionally, our findings highlight the effects of dietary choices on the serum concentrations of these elements. These shifts in elemental concentrations are closely linked to disturbances in antioxidant enzyme function. Such disturbances can increase OS, which may be pivotal in glaucoma development.

The primary approach for treating glaucoma involves interventions for lowering IOP using a combination of topical medications, laser therapy, and surgical procedures. Nevertheless, in certain instances, the disease continues to advance despite successfully lowering the IOP. Consequently, potential non-IOP-related risk factors associated with glaucoma and the mechanisms underlying RGC degeneration have been explored. Previously identified risk factors include genetics, mitochondrial impairment, vascular dysregulation, and OS [13].

OS is characterized by an imbalance between ROS production and antioxidant defenses. ROS, including superoxide anion ($O_2\bullet^-$), hydrogen peroxide ($H_2O_2$), and hydroxyl radical (HO$\bullet$), are byproducts of the mitochondrial cellular metabolism of oxygen. Furthermore, ROS can originate from external sources, including exposure to light radiation, environmental contaminants, xenobiotics, ozone, cigarette smoke, and other factors. Low ROS levels regulate signaling pathways that influence normal physiological and biological reactions [14].

Excessive ROS production decreases adenosine triphosphate synthesis but increases the oxidation of proteins, lipids, and DNA. These mechanisms promote OS, ultimately leading to cell membrane damage, which triggers apoptosis. This process is particularly important in tissues of the central nervous and visual systems, which depend heavily on mitochondrial aerobic energy production. In healthy physiological states, a well-functioning antioxidant system that includes enzymes such as superoxide dismutase and glutathione effectively neutralizes ROS [15].

In the early stages of POAG, reduced blood flow and oxygen delivery to the optic nerve head owing to high IOP have been hypothesized to result in mitochondrial dysfunction, thereby increasing ROS production and promoting OS. These events can trigger inflammatory processes in the eye. This sequence of events and chronic inflammation may damage RGCs, subsequently causing progressive vision loss typically associated with glaucoma [15].

OS has three primary effects on glaucoma. First, it increases IOP by modifying the trabecular meshwork and obstructing the outflow of aqueous humor. Second, it interferes with the self-regulation of blood flow to the optic nerve by affecting the blood vessels. Finally, individuals with diminished antioxidative mechanisms in the eye are at a greater risk of experiencing systemic OS, previously documented as being capable of causing RGC death [13].

In the context of glaucoma development, trace element levels, such as Se, Zn, Cu, manganese, chromium, cobalt, and molybdenum, fluctuate. These elements are crucial in upholding

the equilibrium between pro-oxidative and antioxidative processes owing to their antioxidative properties. They function as cofactors and are found in the prosthetic groups of numerous enzymes [7].

## Selenium

Se, a crucial trace element, is a component of essential antioxidants. Its unique redox properties, associated with selenocysteine, are utilized by antioxidant enzymes such as glutathione peroxidases (GPx), thioredoxin reductases, and iodothyronine deiodinases. GPx is crucial in preserving membrane integrity by reducing the levels of ROS metabolites [16].

Several studies have established a relationship between Se levels and glaucoma development [17, 18]. However, no universally accepted Se level serves as a diagnostic or predictive marker for glaucoma. In a study by Bruhn et al., Se levels in the blood plasma and aqueous humor were compared between individuals with and without POAG. The research by these authors identified an association between a high plasma Se concentration and glaucoma [17]. In contrast to the study by Bruhn et al., male and female patients with glaucoma in our study had significantly lower serum Se levels compared to the control group. The factors contributing to these discrepancies could be racial differences. The patients in Bruhn et al. were predominantly of Hispanic ethnicity, comprising almost 80% of their study population. In contrast, our study consisted entirely of individuals of Caucasian ethnicity. The poorer dietary habits regarding selenium intake in our population compared to those of the Hispanic population in Bruhn et al. may have influenced the study results. Moreover, soils in Poland are considerably deficient in total selenium content. These might be factors contributing to the observed differences between those two studies. The serum Se concentration was determined based on the Se absorbed through daily dietary intake. The serum selenium concentration was influenced by 15 specific dietary habits, which collectively accounted for 50% of the variation. Regular fish, meat, coffee, milk, and egg consumption increased Se concentrations. In line with these findings, Kieliszek and Błażejak demonstrated that food products with high Se content include meat, fish, offal, and cereals [19].

Despite incorporating selenium-rich products into the diet of patients with glaucoma, their serum Se levels remained low. This could be attributed to the variability in Se content, which is influenced by factors such as soil conditions. As mentioned above, the soils in Poland are considerably deficient in total Se content, not exceeding 0.5 mg/kg s.m. For the Podlaskie Voivodeship where the study population resided, it is even only 0.06–0.4 mg/kg s.m. Furthermore, research has been conducted in Poland since the 1980s to determine the content of this element in biological samples (blood serum and urine), with tests being conducted approximately every 10 years. From 1981 to 1999, a two-fold decrease in serum Se concentration was observed in all age groups that underwent the tests. For instance, a decrease in blood serum Se concentration was observed in non-pregnant women, decreasing from 95 ± 13 μg/L in 1981 to 54 ± 12 μg/L in 1999. However, the standard range of Se concentration for adults is optimal for the activity of serum GPx, is 66–104 μg/L. We hypothesize that the previously observed decrease in Se bioavailability in our study area may have led to a consequent decrease in Se levels in the agricultural and animal products consumed by our study population.

Se absorption from foods is also influenced by dietary elements, such as fat, protein, and the presence of heavy metals [20]. Moreover, consuming beer, beverages, lard, white bread, and potatoes leads to decreased serum Se levels.

Numerous studies have described the serum levels of Se in smokers and non-smokers. Luty-Frackiewicz et al. [21] described the dietary habits of the population in Silesia and observed a decrease in Se levels in the serum of smokers compared to that of non-smokers;

however, these authors did not observe a significant difference between these groups. Some reports described equal serum Se levels in smoking and non-smoking groups [16, 22], whereas other studies reported significant group differences [17, 23]. In our study, a significant difference in Se levels between smokers and non-smokers was observed. This can be explained based on the work of Luty-Frackiewicz et al., who suggested that smokers may require an increased intake of vitamins and nutrients by approximately 40%, which could influence the significant deficiencies in Se levels [21]. This in turn can be explained by the greater OS in smokers caused by increased exposure to toxic elements (such as Cd and Pb) due to smoking cigarettes. Selenium has a detoxifying effect when exposed to toxic elements and may be washed out faster in comparison to non-smokers.

## Zinc

Zn, an essential micronutrient, is crucially involved in the active sites of approximately 300 enzymes and is integral to numerous signaling pathways. It is widely recognized for its antioxidant properties and its role in eliminating superoxide radicals [7].

In our study, we observed lower serum concentrations of Zn in female and male participants with glaucoma than in healthy controls. A decrease in the serum concentration of Zn can disrupt antioxidant defense mechanisms and its ability to maintain cellular integrity, resulting in increased OS, which is considered to contribute to glaucoma development. Akyol et al. examined Zn and Cu levels in the serum and aqueous humor of individuals with glaucoma and cataracts. The glaucoma group had the highest Cu concentration, and a notable negative correlation was observed between the Zn and Cu levels in the serum of patients with glaucoma. This suggests that elevated Cu and reduced Zn levels may be significant factors in patients with glaucoma [24].

According to our research findings, 15 nutritional factors collectively accounted for 42% of the variation in serum Zn concentration. Notably, the regular intake of ham, offal, and potatoes has been positively associated with higher serum Zn levels. Meat products are a good source of Zn because they contain complete proteins that can increase Zn bioavailability. Conversely, the consumption of vegetable oil and raw vegetables may be linked to a significant decrease in serum Zn concentration because plant products contain dietary fiber and phytates, which reduce Zn bioavailability.

## Copper

Cu is typically acknowledged for its antioxidant functions in human metabolism and is essential in several critical processes, such as ROS detoxification, energy metabolism, Fe absorption, and cellular signaling. The biological role of Cu is intricately linked to its participation in the structures and activities of various enzymes, including Cu/Zn superoxide dismutase, β-dopamine hydroxylase, cytochrome c, tyrosinase, ceruloplasmin, hephaestin, and lysyl oxidase. Cu Zn superoxide dismutase demonstrates antioxidant capabilities by effectively neutralizing superoxide radicals and is crucial in controlling apoptotic signaling and mitigating OS [7, 25].

In our study, the serum Cu concentration of patients with glaucoma did not vary significantly from that of the control group. However, the glaucoma group exhibited an elevated Cu/Zn ratio compared to the control group. A high Cu/Zn ratio has been observed in older adults, particularly in those with neurodegenerative disorders, including glaucoma. Furthermore, an elevated Cu/Zn ratio is associated with an increased risk of cardiovascular-related mortality, malignancy, and all-cause mortality in older adults [26].

We observed a significant decrease in serum Cu levels in male and female patients with glaucoma compared to sex-matched healthy controls. Numerous reports have suggested a protective role of Cu as a metabolic antioxidant.

## TAS

OS can be evaluated using various biomarkers, and multiple methods have been developed to measure total oxidative capacity by assessing each antioxidant component individually. TAS is a comprehensive measure aimed at describing the balance between pro-oxidants and antioxidants in the plasma. Studies have revealed reduced serum TAS levels in patients with glaucoma [27].

In the current study, we did not observe significant differences in serum TAS levels of patients with glaucoma compared to those of participants in the control group. Therefore, dietary habits may have limited direct effects on serum TAS levels. We included 12 dietary factors in the regression model, and only frequent consumption of margarine had a significant negative effect, leading to a decrease in serum TAS levels. Consuming saturated fat-rich foods, such as margarine, can contribute to the onset of metabolic complications. This includes an increased production of ROS, which results in increased OS [28].

## Pathophysiological differences between POAG, PEXG, and PACG

Glaucoma is a disease of multifactorial etiology. Under physiological conditions, aqueous humor flows from the posterior to the anterior eye chamber and is then drained via outflow pathways including the trabecular meshwork, choroidal-scleral pathway, and the iris. In POAG, the outflow of the aqueous humor is hindered due to increased resistance at the level of these outflow tracts. One of the most common causes of secondary open-angle glaucoma is PEXG, associated with the deposition of protein-like material within the anterior segment of the eye, especially in the iridocorneal angle and lens capsule. In PACG, the peripheral part of the iris overlaps the trabecular meshwork, blocking the outflow of the aqueous humor. Although, we hypothesized possible effects of trace element concentrations on specific glaucoma types, our study revealed no significant differences.

## Conclusions

The studied microelements may be pivotal in modulating the activity of critical antioxidant enzymes associated with glaucoma, influencing various processes, including programmed cell death, inflammatory responses, OS, and neuroprotection. Dietary pattern modification can influence the levels of serum elements, thus providing opportunities to prevent the occurrence of glaucoma in high-risk patients. We emphasize that examining serum levels of microelements can provide valuable insights into the biochemical profile of glaucomatous eye changes, offering new low-cost opportunities for early prevention and functional, supportive treatments. The results obtained in this project may help provide better advice on dietary adjustment, thereby supporting treatment and slowing glaucoma progression. We believe that this study serves as a basis for considering personalized nutritional therapy for the prevention and supportive treatment of patients with early-stage glaucoma. However, the aforementioned results should be further investigated in a significantly larger study population and tailored to region-specific dietary habits. Further studies are needed to identify the cutoff values of microelement concentrations that distinguish between various types of glaucoma and their potential as pharmacotherapy effectiveness markers.

## Supporting information

**S1 Table. Relationships between frequency of food product consumption and serum Se, Cu, Zn, and TAS levels according to multiple stepwise regression analysis.**
(DOCX)

**S1 File. Certificate of English language editing.**
(PDF)

**S2 File. Dataset used to statistical analysis.**
(DOCX)

## Acknowledgments

We would like to thank Editage for English language editing (S1 File).

## Author Contributions

**Conceptualization:** Katarzyna Socha, Joanna Konopińska.

**Data curation:** Izabela Zawadzka, Maryla Młynarczyk, Martyna Falkowska, Katarzyna Socha, Joanna Konopińska.

**Formal analysis:** Izabela Zawadzka, Maryla Młynarczyk, Martyna Falkowska, Katarzyna Socha, Joanna Konopińska.

**Investigation:** Izabela Zawadzka, Maryla Młynarczyk, Joanna Konopińska.

**Methodology:** Katarzyna Socha, Joanna Konopińska.

**Project administration:** Joanna Konopińska.

**Resources:** Katarzyna Socha.

**Supervision:** Katarzyna Socha, Joanna Konopińska.

**Validation:** Joanna Konopińska.

**Writing – original draft:** Izabela Zawadzka.

**Writing – review & editing:** Katarzyna Socha, Joanna Konopińska.

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
