## [Decision Letter · Decision Letter 0]

6 Feb 2024

PONE-D-23-43349Dietary patterns; serum concentrations of selenium, copper, and zinc; Cu/Zn ratio; and total antioxidant status in patients with glaucomaPLOS ONE

Dear Dr. Konopińska,

Thank you for submitting your manuscript to PLOS ONE. After careful consideration, we feel that it has merit but does not fully meet PLOS ONE’s publication criteria as it currently stands. Therefore, we invite you to submit a revised version of the manuscript that addresses the points raised during the review process.

Thank you for submitting the following manuscript to PLOS ONE.Please revise the manuscript according to the reviewers' comments and upload the revised file.

We look forward to receiving your revised manuscript.

Kind regards,

Yung-Hsiang Chen, Ph.D.

Academic Editor

PLOS ONE

Additional Editor Comments:

Thank you for submitting the following manuscript to PLOS ONE.

Please revise the manuscript according to the reviewers' comments and upload the revised file.

Reviewers' comments:

Reviewer's Responses to Questions

**Comments to the Author**

1. Is the manuscript technically sound, and do the data support the conclusions?

Reviewer #1: Yes

Reviewer #2: Yes

Reviewer #3: Yes

2. Has the statistical analysis been performed appropriately and rigorously? 

Reviewer #1: Yes

Reviewer #2: Yes

Reviewer #3: Yes

3. Have the authors made all data underlying the findings in their manuscript fully available?

Reviewer #1: Yes

Reviewer #2: Yes

Reviewer #3: Yes

4. Is the manuscript presented in an intelligible fashion and written in standard English?

Reviewer #1: Yes

Reviewer #2: Yes

Reviewer #3: Yes

5. Review Comments to the Author

Reviewer #1: Abstract

I think there is a typographic error in the abstract. Did you measure the concentration of the trace elements in the aqueous humor?

patients with glaucoma had higher Se levels?

Introduction

Glaucoma is the FIRST leading cause of irreversible blindness.

The only MODIFIABLE risk factor for glaucoma is elevated IOP.

Provide references for lines 67-70.

Line 81: The concentration of the trace elements does not essentially determine the occurrence of glaucoma. Also, this should be corrected in the abstract.

Methods

Use corrected distance visual acuity instead of BCVA.

Discussion

Line 338-339: How do you justify the discrepancy between your study and Bruhn et al. study?

I recommend a detailed discussion on the pathophysiological differences between POAG, PXG, and PACG and also the impact of the concertation of the trace elements on the specific type of glaucoma.

Reviewer #2: The work ´Dietary patterns; serum concentrations of selenium, copper, and zinc; Cu/Zn ratio; and total antioxidant status in patients with glaucoma` aimed to identify the biochemical parameters that determine the occurrence of glaucoma and to assess the correlation between oxidative stress and clinical data in patients with glaucoma and healthy controls. It provides insights on the glaucoma population and the amount of individuals enrolled is relevant. However, some improvements of the manuscript are required for its publication.

- The goal of the manuscript seems to be ambitious, as 3 trace elements and total oxidant status were used to identify biochemical parameters that determine the occurrence of glaucoma.

- Introduction

Line 50. ´This indicates the presence of undefined molecular mechanisms underlying glaucoma development´. Is this applicable to onset and progression of glaucoma?

Some references must be added along the introduction:

Lines 65-67. Reference on aqueous humor.

Line 70. Reference.

Line 76. Reference.

- Line 83. A comparison with other, at least European, data in serum of healthy and glaucoma patients may be added (if possible) in the results/discussion of the manuscript.

-Inclusion and exclusion criteria. Line 112. Was the presence of cataracts in the healthy population considered?

- Mineral component and total antioxidant status determination. Line 154. The obtained and certified values must be added to the manuscript with the subsequent statistical evaluation. Indicate how is expressed the precision (RSD?)

- Line 159. Please, rephrase the sentence. It is not clear what is compared. Reference values correspond to…

- Demographic characteristics of the study groups. Table 1 is not referenced in the text.

- Lines 189-190. The age must be indicated with three significant digits (one number after comma)

- Line 198. ´Study and control group´. Replace study by glaucoma group.

- Line 205. Please, clarify or define r

- Lines 224-226. Please, revise grammatically this sentence.

- Line 251. Can be this patient an outlier? If so, could it be excluded for the statistical comparison?

- Line 268. Does this group of individuals have lower consumption of fish? This should be considered as it is a major source of Se compared to the other food products.

- Table 6. It can be included in the supporting information.

- Lines 333-334. Please, add reference

- Lines 341-342. Please, rephrase the sentence.

- Line 353. Please, harmonize the units of the concentrations.

As a general remark, was the comparison/evaluation of the food consumption performed by separate groups (glaucoma group and healthy group)? Was any difference found?

Line 344. Was the diet of the glaucoma patients enriched in Se? This is not clear along the manuscript (section of inclusion/exclusion criteria and discussion).

Reviewer #3: This study is very interseting and suitable for publication

Only English edition should be done before publication

This study should be effect the treatment of glaucoma and this condition is very important for glaucoma progression

6. PLOS authors have the option to publish the peer review history of their article (what does this mean?). If published, this will include your full peer review and any attached files.

Reviewer #1: No

Reviewer #2: No

Reviewer #3: No

---

## [Author Response · Author response to Decision Letter 0]

3 Mar 2024

Responses to the Editorial Board Members’ comments

March 03 2024

Dear Editorial Board Members and Reviewers,

We would like to thank you for the detailed review of our manuscript and your valuable remarks. The manuscript has been rechecked, and the necessary changes have been made in accordance with the Reviewers’ suggestions. The responses to all comments have been prepared and are provided below. We hope that you will find our explanations and manuscript modifications satisfactory and consider the revised manuscript for publication in PLoS One.

Reviewer 1.

Comment 1: Abstract

I think there is a typographic error in the abstract. Did you measure the concentration of the trace elements in the aqueous humor?

Response: We appreciate your attention to detail; indeed, it is a typographical error. We analyzed the levels of trace elements (selenium, copper, zinc, and Cu/Zn ratio) and total antioxidant status (TAS) in blood serum samples. Following your recommendation, we have incorporated the necessary corrections into the article. 

Comment 2: patients with glaucoma had higher Se levels?

Response: This is another typographical error, in fact, the total Se serum levels are lower in patients with glaucoma than in control participants. We have corrected this mistake. 

Comment 3: Introduction

Glaucoma is the FIRST leading cause of irreversible blindness.

Response: Thank you for pointing this out, we corrected it. 

Comment 4: The only MODIFIABLE risk factor for glaucoma is elevated IOP.

Response: Thank you for pointing this out, we corrected it. 

Comment 5: Provide references for lines 67-70.

Response: Thank you for this comment. We have added the appropriate references which can be found as references number 7 and 8 in the reference list.

Comment 6: Line 81:The concentration of the trace elements does not essentially determine the occurrence of glaucoma. Also, this should be corrected in the abstract.

Response: We appreciate this pertinent comment. We have changed these statements in the manuscript. 

Comment 7: Methods

Use corrected distance visual acuity instead of BCVA.

Response: Thank you for this suggestion. We have changed the terminology in the entire manuscript. 

Comment 8: Discussion

Line 338-339: How do you justify the discrepancy between your study and Bruhn et al. study?

Response: Thank you for this insightful comment. We believe that one of the factors contributing to the discrepancies between these two studies could be racial differences. The participants in Bruhn et al. were predominantly of Hispanic ethnicity, comprising almost 80% of their study population. In contrast, our study population consisted entirely of individuals of Caucasian ethnicity. Additionally, the poor dietary habits regarding selenium intake in our population may have significantly influenced the results when compared to the Hispanic study group in Bruhn et al. Moreover, as indicated on lines 376-387, soils in Poland, particularly in our region, are considerably deficient in total selenium content. These might be the factors that determine the differences between those two papers. We have added an appropriate explanation to the Discussion section (lines 405-411).

Comment 9: I recommend a detailed discussion on the pathophysiological differences between POAG, PXG, and PACG and also the impact of the concertation of the trace elements on the specific type of glaucoma.

Response: Thank you for your suggestion. In fact, we conducted an analysis comparing the concentration of trace elements in various types of glaucoma. However, our study revealed no significant differences. Therefore, we opted not to discuss this topic further. Following your recommendation, we have added Table 6 presenting the results from our statistical analysis, as well as a chapter in the Discussion section as follows (lines 462-472):

Pathophysiological differences between POAG, PEXG, and PACG 

Glaucoma is a disease of multifactorial etiology. Under physiological conditions, aqueous humor flows from the posterior to the anterior eye chamber and is then drained via outflow pathways including the trabecular meshwork, choroidal-scleral pathway, and the iris. In POAG, the outflow of the aqueous humor is hindered due to increased resistance at the level of these outflow tracts. One of the most common causes of secondary open-angle glaucoma is PEXG, associated with the deposition of protein-like material within the anterior segment of the eye, especially in the iridocorneal angle and lens capsule. In PACG, the peripheral part of the iris overlaps the trabecular meshwork, blocking the outflow of the aqueous humor. Although, we hypothesized possible effects of trace element concentrations on specific glaucoma types, our study revealed no significant differences.

Table 6. Comparison of Se, Zn, Cu, and TAS levels and Cu/Zn ratio between glaucoma types.

Variable POAG (n = 58) PEXG (n = 16) PACG (n = 17) Other types (n = 7) p

 Mean ± SD Median (Q1; Q3) Mean ± SD Median (Q1; Q3) Mean ± SD Median (Q1; Q3) Mean ± SD Median (Q1; Q3) 

Se, mg/L 66.44 ± 17.37 63.74 (55.42; 75.97) 76.40 ± 18.53 71.05 (63.23; 81.46) 72.73 ± 17.14 72.81 (59.88; 83.02) 67.63 ± 11.38 66.91 (59.78; 76.19) 0.175

Zn, mg/L 0.77 ± 0.14 0.76 (0.68; 0.85) 0.74 ± 0.09 0.77 (0.70; 0.81) 0.76 ± 0.09 0.76 (0.71; 0.82) 0.77 ± 0.11 0.75 (0.71; 0.82) 0.912

Cu, mg/L 1.05 ± 0.36 1.02 (0.83; 1.28) 1.22 ± 0.32 1.17 (1.05; 1.45) 1.03 ± 0.28 1.07 (0.92; 1.17) 1.17 ± 0.30 1.16 (0.99; 1.26) 0.281

Cu/Zn, molar ratio 1.45 ± 0.55 1.38 (1.02; 1.79) 1.72 ± 0.56 1.61 (1.38; 1.94) 1.42 ± 0.41 1.47 (1.20; 1.71) 1.56 ± 0.35 1.67 (1.28; 1.79) 0.263

TAS, mmol/L 1.53 ± 0.25 1.52 (1.40; 1.65) 1.53 ± 0.28 1.55 (1.42; 1.64) 1.49 ± 0.35 1.58 (1.43; 1.64) 1.51 ± 0.17 1.50 (1.45; 1.65) 0.9771

Notes: POAG; PEXG, ; PACG, ; Q1, first quartile; Q3, third quartile; SD, standard deviation. Data were compared with ANOVA or Kruskal–Wallis test1.

Reviewer #2: 

Comment 1: Introduction

Line 50. ´This indicates the presence of undefined molecular mechanisms underlying glaucoma development´. Is this applicable to onset and progression of glaucoma?

Response: Thank you for this pertinent comment. Accordingly, we have slightly changed this sentence as follows:

“This might indicate the presence of undefined molecular mechanisms underlying glaucoma development.”

Comment 2: Some references must be added along the introduction:

Lines 65-67. Reference on aqueous humor, and line 70. Reference.

Response: Thank you for bringing attention to missing references. We added the following reference at line 70.

Comment 3: Line 76. Reference. 

Response: Thank you for this comment, we added an appropriate reference which can be found as reference 10 in the bibliography. 

Comment 4: Line 83. A comparison with other, at least European, data in serum of healthy and glaucoma patients may be added (if possible) in the results/discussion of the manuscript.

Response: Thank you for your suggestion. During our search in medical databases, we did not find similar studies to ours. Most publications either evaluated only the concentrations of trace elements in serum or aqueous humor without assessing the potential role of oxidative stress or assessed the role of oxidative stress without connecting it to the levels of trace elements. In our work, we compare trace elements that regulate oxidative stress, which can influence the development of glaucoma. Hence, we cannot compare those papers with our study. 

Comment 5: Inclusion and exclusion criteria. Line 112. Was the presence of cataracts in the healthy population considered?

Response: Thank you for this question. Yes, patients with cataract were excluded from the study. We have added this information to the Methods section. 

Comment 6: Mineral component and total antioxidant status determination. Line 154. The obtained and certified values must be added to the manuscript with the subsequent statistical evaluation. Indicate how is expressed the precision (RSD?)

Response: In response to your question, we have added the results of the method checks to the table. Precision is in this study expressed as a compactness factor in %, which is SD divided by the value for the six individual materials of the certificate multiplied by 100.

Comment 7: Line 159. Please, rephrase the sentence. It is not clear what is compared. Reference values correspond to…

Response: Thank you for your comment. We have rephrased the sentence as follows: “The serum concentrations of Se, Cu, and Zn in the glaucoma group were subsequently evaluated in comparison to the reference values of healthy controls (66–104 µg/L, 0.7–1.6 mg/L, and 0.7–1.3 mg/L, respectively) [11].” (lines 191-193).

Comment 7: Demographic characteristics of the study groups. Table 1 is not referenced in the text.

Response: Thank you for this comment. Table 1 is now referenced on line 224. 

Comment 8: Lines 189-190. The age must be indicated with three significant digits (one number after comma)

Response: Thank you for pointing this out. We have corrected our data reporting.

Comment 9: Line 198. ´Study and control group´. Replace study by glaucoma group.

Response: Thank you for this suggestion. We fully agree with your comment and have corrected it in the entire manuscript.

Comment 10: Line 205. Please, clarify or define r

Response: Spearman’s rank test correlation coefficient allows to determine whether there is a linear relationship between two variables, and if so, it allows us to determine its strength and its direction, i.e., whether the correlation is positive or negative. 

Comment 11: Lines 224-226. Please, revise grammatically this sentence. 

Response: Thank you for this comment. We have rephrased the sentence as follows:

“Significantly lower serum levels of Se and Zn were observed in men (67.7 ± 17.14 μg/L and 0.76 ± 0.11 mg/L, respectively) and women (68.73 ± 16.21 μg/L and 0.76 ± 0.13 mg/L, respectively) diagnosed with glaucoma than in healthy men (78.64 ± 15.13 μg/L and 0.82 ± 0.21 mg/L, respectively) and women (88.23 ± 56.43 μg/L and 0.71 ± 0.14 mg/L, respectively).” (lines 236-240). 

Comment 12: Line 251. Can be this patient an outlier? If so, could it be excluded for the statistical comparison?

Response: Thank you for this comment. We excluded the patient from the study.

Comment 13: Line 268. Does this group of individuals have lower consumption of fish? This should be considered as it is a major source of Se compared to the other food products.

Response: Thank you for your insightful comment. We agree with your point and added the following explanation to the Methods section: “For this study, consumption was frequent if specific food products were consumed twice to thrice weekly or more, except for fish and fish products, for which consumption once or more weekly was considered frequent. Food items consumed less frequently were categorized as those consumed once or more weekly. This categorization aligns with the recommendations of the Committee on Human Nutrition Science of the Polish Academy of Sciences for assessing the frequency of consuming specific food products.” (lines 131-137).

Moreover, we added these data to the Results section: “In the glaucoma group, the effect of fish consumption as a source of selenium was statistically significant; patients who consumed fish more frequently had higher selenium levels compared to those who consumed fish less frequently (without considering other dietary factors).“ (lines 301-303).

We also added Table 8 as follows:

Table 8. Relationship between the frequency of fish consumption and the serum concentration of selenium in patients with glaucoma.

Fish consumption frequency Number of patients with glaucoma Concentration of selenium in serum (µg/L) p

Frequent (more than once a week) 51 74.92 ± 17.49 < 0.005*

Seldom (less or equal to once a week) 53 64.87 ± 13.40 

Comment 14: Table 6. It can be included in the supporting information.

Response: Thank you for this remark. We have moved this table to the Supplementary data as the S1 Table. 

Comment 15: Lines 333-334. Please, add reference 

Response: Thank you for this comment. Following your suggestion, we have added an appropriate reference.

Comment 16: Lines 341-342. Please, rephrase the sentence.

Response: Thank you for that remark. We have rephrased the sentence. 

Comment 16: Line 353. Please, harmonize the units of the concentrations.

Response: Thank you for pointing this out. Accordingly, we corrected the units throughout our manuscript. 

Comment 17: As a general remark, was the comparison/evaluation of the food consumption performed by separate groups (glaucoma group and healthy group)? Was any difference found?

Response: Thank you for this remark. Unfortunately, this statistical analysis was only performed in the glaucoma group, so we cannot compare the groups. 

Comment 18: Line 344. Was the diet of the glaucoma patients enriched in Se? This is not clear along the manuscript (section of inclusion/exclusion criteria and discussion).

Response: In response to your comment, we have carefully revised the Methods section to include a more detailed explanation about excluding patients with any pharmacological supplementation: “Additionally, individuals taking dietary vitamin supplements, especially those containing Se, Zn, or Cu, or medications that could potentially affect the trace element levels were excluded from the study.” (lines 117-120).

Reviewer #3: 

Comment 1: This study is very interesting and suitable for publication

Only English edition should be done before publication

This study should be effect the treatment of glaucoma and this condition is very important for glaucoma progression

Response: Thank you for your encouraging comments and the constructive feedback regarding the quality of the data presentation in our manuscript. We have revised the entire manuscript as advised. We have add the English editing certificate as a supplementary file.

---

## [Decision Letter · Decision Letter 1]

19 Mar 2024

Dietary patterns; serum concentrations of selenium, copper, and zinc; copper/zinc ratio; and total antioxidant status in patients with glaucoma

PONE-D-23-43349R1

Dear Dr. Konopińska,

We’re pleased to inform you that your manuscript has been judged scientifically suitable for publication and will be formally accepted for publication once it meets all outstanding technical requirements.

Kind regards,

Yung-Hsiang Chen, Ph.D.

Academic Editor

PLOS ONE

Additional Editor Comments (optional):

Congratulations on the acceptance of your manuscript, and thank you for your interest in submitting your work to PLOS ONE.

Reviewers' comments:

Reviewer's Responses to Questions

**Comments to the Author**

1. If the authors have adequately addressed your comments raised in a previous round of review and you feel that this manuscript is now acceptable for publication, you may indicate that here to bypass the “Comments to the Author” section, enter your conflict of interest statement in the “Confidential to Editor” section, and submit your "Accept" recommendation.

Reviewer #2: All comments have been addressed

2. Is the manuscript technically sound, and do the data support the conclusions?

Reviewer #2: Yes

3. Has the statistical analysis been performed appropriately and rigorously? 

Reviewer #2: Yes

4. Have the authors made all data underlying the findings in their manuscript fully available?

Reviewer #2: Yes

5. Is the manuscript presented in an intelligible fashion and written in standard English?

Reviewer #2: Yes

6. Review Comments to the Author

Reviewer #2: As a small remark and following a previous comment, it is convenient that the experimental data obtained for the reference material (Seronorm serum) will be added to the manuscript, and also to indicate in the manuscript how the precision is expressed.

7. PLOS authors have the option to publish the peer review history of their article (what does this mean?). If published, this will include your full peer review and any attached files.

Reviewer #2: No

---

## [Editor Report · Acceptance letter]

22 Mar 2024

PONE-D-23-43349R1 

PLOS ONE

Dear Dr. Konopińska, 

I'm pleased to inform you that your manuscript has been deemed suitable for publication in PLOS ONE. Congratulations! Your manuscript is now being handed over to our production team.

Kind regards, 

on behalf of

Dr. Yung-Hsiang Chen 

Academic Editor

PLOS ONE